# Hybridization of Nanodiamond and CuFe-LDH as Heterogeneous Photoactivator for Visible-Light Driven Photo-Fenton Reaction: Photocatalytic Activity and Mechanism

**Lu Liu [1], Shijun Li [2], Yonglei An [2,\*], Xiaochen Sun [1], Honglin Wu [2], Junzhi Li [1], Xue Chen [2] and Hongdong Li [1,\*]**

[1] State Key Laboratory of Superhard Materials, College of Physics, Jilin University, Changchun 130012, China; liulu15@mails.jlu.edu.cn (L.L.); xcsun17@mails.jlu.edu.cn (X.S.); lijz17@mails.jlu.edu.cn (J.L.)

[2] Key Laboratory of Groundwater Resources and Environment (Jilin University), Ministry of Education, Changchun 130021, China; lisj2516@mails.jlu.edu.cn (S.L.); wuhl2516@mails.jlu.edu.cn (H.W.); chenxue2515@mails.jlu.edu.cn (X.C.)

[\*] Correspondence: anyonglei85@jlu.edu.cn (Y.A.); hdli@jlu.edu.cn (H.L.)

**Abstract:** Establishing a heterojunction for two kinds of semiconductor catalysts is a promising way to enhance photocatalytic activity. In this study, nanodiamond (ND) and CuFe-layered double hydroxide (LDH) were hybridized by a simple coprecipitation method as a novel heterojunction to photoactivate $H_2O_2$. The ND/LDH possessed a hydrotalcite-like structure, large specific surface area ($S_{BET}$ = 99.16 $m^2$/g), strong absorption of visible-light and low band gap ($E_g$ = 0.94 eV). Under the conditions of ND/LDH dosage 0.0667 g/L, $H_2O_2$ concentration 19.6 mmol/L, and without initial pH adjustment, 93.5% of 10 mg/L methylene blue (MB) was degraded within 120 min, while only 78.3% of MB was degraded in the presence of LDH instead of ND/LDH. The ND/LDH exhibited excellent stability and maintained relatively high activity, sufficient to photoactivate $H_2O_2$ even after five recycles. The mechanism study revealed that in the heterojunction of ND/LDH, the photoelectrons transferred from the valence band of LDH (Cu/Fe 3d $t_{2g}$) to the conduction band of LDH (Cu/Fe 3d $e_g$) could spontaneously migrate onto the conduction band of ND, promoting the separation of photo-induced charges. Thus, the photoelectrons had sufficient time to accelerate the redox cycles of $Cu^{3+}$/$Cu^{2+}$ and $Fe^{3+}$/$Fe^{2+}$ to photoactivate $H_2O_2$ to produce hydroxyl radicals, resulting in excellent photo-Fenton efficiency on MB degradation.

**Keywords:** heterojunction; layered double hydroxide; nanodiamond; photo-Fenton

## 1. Introduction

Water pollution with synthetic organics (e.g., dyes, pesticides, pharmaceuticals) is a worldwide problem because these unnatural organic pollutants are difficult to degrade with traditional chemical or biochemical processes [1,2]. Fortunately, advanced oxidation processes (AOPs) possess excellent oxidizing abilities for decomposing refractory organic pollutants, due to the oxidation of highly active radicals such as hydroxyl radical (HO·) [3–6]. As a typical AOP, Fenton technology is a simple and effective method for treating organic wastewater [7,8]. However, the leaching problem of catalysts (e.g., copper ion, iron ion) limits the application of Fenton technology [7–10]. In addition, the Fenton assisted by photoirradiation (photo-Fenton) is more efficient than the traditional Fenton [11,12]. Therefore, improving the utilization efficiency of visible-light is vital for the photo-Fenton because it can make full use of solar-light in which visible-light is absolutely dominant [3,13].

In recent years, layered double hydroxide (LDH) has been widely used as a catalyst carrier, or precursor, owing to its specific physicochemical properties which arises from its exchangeable metal cations and interlayer anions [14,15]. In general, LDH possesses such low band-gap energy ($E_g$) that LDH can be easily excited by UV-Vis light. Thus, the photoelectrons of LDH would be captured by other chemicals to participate in reduction reactions, for example, water reduction and photocatalytic degradation [16–18]. More importantly, LDH can be reused as a heterogeneous catalyst and will not cause catalyst leakage and secondary metallic pollution [19,20]. In addition, copper ions and iron ions are two typical and efficient active-species for catalyzing the Fenton reaction [7,21]. Therefore, we set out to synthesize a stable and efficient LDH catalyst through doping $Cu^{2+}$ and $Fe^{3+}$ to catalyze the photo-Fenton reaction because there have been no reports of utilizing CuFe-LDH as a photo-Fenton catalyst.

As is commonly known, co-catalysis and element doping are both efficient ways of enhancing the photocatalytic activity of semiconductor photocatalysts [22]. Typically, co-catalysis is conducted through establishing a heterojunction which is effective for promoting the separation of photoexcitons (electron and hole) [22]. In recent years, nanodiamond (ND) has often been hybridized with other semiconductor photocatalysts to form heterojunctions for enhancing photocatalysis, due to its special physicochemical and optical properties and harmlessness to living organisms [23–26]. For example, the hybridization of ND and $TiO_2$ was used to photodegrade organic pollutants [27,28], the hybridization of ND and $Cu_2O$ was used for photocatalytic hydrogen evolution [23], and the hybridization of ND and gold was used to photocatalyze the Fenton reaction [29–31]. Indeed, the unrivalled carrier mobility and large specific surface area of ND may facilitate the transfer of photogenerated carriers to the photocatalyst surface [23,32]. However, these reactions were all conducted under UV-light or solar-light (~7% is UV-light) irradiation [1]. If the photocatalysis on a ND heterojunction can be effectively driven by visible-light, it would be of great significance for the improvement of solar energy utilization.

Herein, a novel heterojunction hybridized by ND and CuFe-LDH was synthesized as a high-efficiency heterogeneous photo-Fenton catalyst. The crystal structure was characterized by high-resolution transmission electron microscopy (HRTEM) and X-ray diffraction (XRD), and the electronic structure was analyzed by X-ray photoelectron spectra (XPS) and UV-Vis diffuse reflectance spectra (DRS). Furthermore, the photocatalytic performance and mechanism of ND/LDH in the photo-Fenton reaction system were investigated in detail in this study.

## 2. Results and Discussion

### 2.1. Characterization of ND/LDH

HRTEM images of LDH and ND/LDH are shown in Figure 1. Four kinds of lattice fringes for bare LDH were clearly found in Figure 1B; the lattice fringe spacings were 0.243, 0.233, 0.217 and 0.136 nm, corresponding to (104), (015) crystal planes of the LDH phase referred to on PDF card No. 89-461 and (114), (209) crystal planes of LDH phase referred to on PDF card No. 87-1138, respectively. For ND/LDH, four typical lattice fringes were also clearly found in Figure 1D; the lattice fringe spacings were 0.243, 0.233, 0.205 and 0.136 nm, with 0.205 nm corresponding to the (111) crystal plane of cubic diamonds [33]. These results suggested that the as-prepared ND/LDH possessed a hydrotalcite-like structure, and that the ND was hybridized with LDH successfully. Moreover, the clear lattice fringes indicated good crystallinity of ND/LDH.

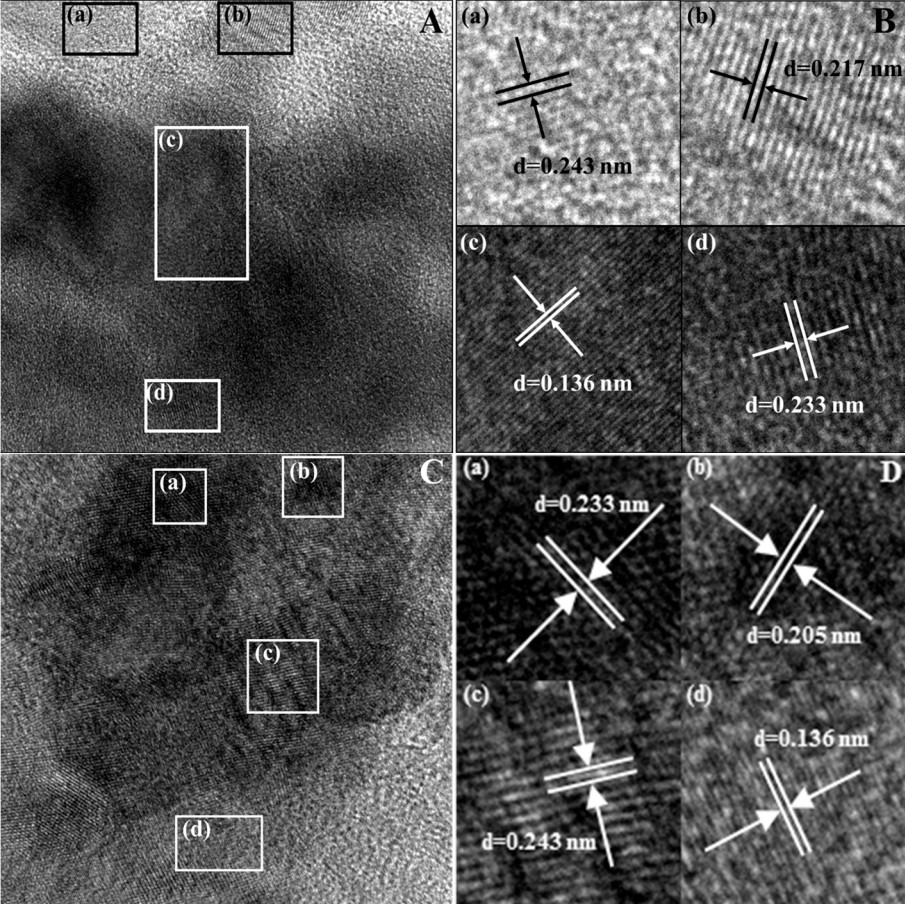

**Figure 1.** High-resolution transmission electron microscopy (HRTEM) images of layered double hydroxide (LDH) (**A**,**B**) and nanodiamond/layered double hydroxide (ND/LDH) (**C**,**D**). (**a**–**d**) are the four kinds of selected lattice fringes for LDH and ND/LDH, respectively.

The crystal phase and crystallinity of ND, LDH and ND/LDH were characterized using an X-ray diffractometer with Cu Kα radiation over a range of 2θ angles from 20 to 90° (Figure 2), respectively. Two representative diffraction peaks located at 44.66 and 78.08° were observed in the ND spectrum, corresponding to the (111) and (220) crystal planes of diamond [33,34]. Two other diffraction peaks located at 38.62 and 64.98° were also observed, corresponding to the (021) and (311) crystal planes of graphite phase referred to in PDF card No. 89-8488, and indicative of partial graphitization on the surface of ND. In the LDH spectrum, the 2θ of diffraction peaks were located at 32.10, 33.28, 35.90, 39.24, 44.66, 64.98 and 78.08°, corresponding to the (101), (009), (012), (015), (018), (116) and (1016) crystal planes of the LDH phase referred to in PDF card No. 89-461 and in the literature [35–37]. The remaining two outstanding diffraction peaks were located at 29.62 and 48.58°, corresponding to (005) and (107) crystal planes of the LDH phase referred to in PDF card No. 87-1138. For ND/LDH, all the diffraction peaks of ND and LDH were observed correspondingly, whereas the diffraction peaks of ND coincided with the partial peaks of LDH. In addition, a new diffraction peak at 22.96° corresponding to the (006) crystal plane of LDH phase was detected in the ND/LDH spectrum [35–37]. These results adequately confirmed the hydrotalcite-like structure of ND/LDH and the successful hybridization of ND and LDH. Moreover, the sharp diffraction peaks indicated good crystallinity of ND/LDH.

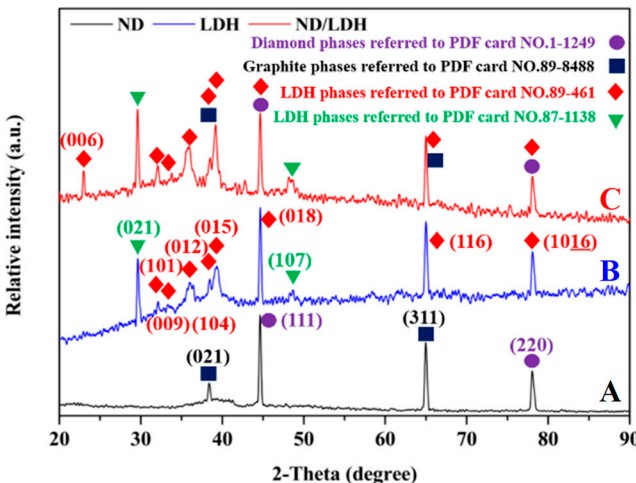

**Figure 2.** X-ray diffraction (XRD) spectra of ND (**A**), LDH (**B**) and ND/LDH (**C**).

The surface state of ND/LDH was investigated with the XPS method. The full-range XPS spectra is represented in Figure 3a, where peaks corresponding to C 1s, Cu 2p and Fe 2p were detected. The sharp peak of C 1s was composed of four splitting peaks at 284.39, 284.96, 286.06 and 287.83 eV (Figure 3b). 284.96 eV suggested sp3-hybridized carbon, indicating the existence of ND. 284.39 eV suggested sp2-hybridized carbon, indicating partial graphitization on the surface of ND, which was consistent with the XRD results. 286.06 and 287.83 eV suggested C-O and C=O structures, respectively, attributed to the oxygen-containing groups on the surface of ND [38]. For Cu 2p (Figure 3c), two typical peaks were located at 954.62 and 934.64 eV, corresponding to Cu $2p_{1/2}$ and Cu $2p_{3/2}$. Correspondingly, the other two peaks, at 942.68 and 962.75 eV, were the satellite peaks of Cu 2p. These results suggest that the copper in ND/LDH appeared as $Cu^{2+}$ with the outermost electron configuration of $3d^9$ [39–42]. For Fe 2p (Figure 3d), the XPS peak of Fe $2p_{1/2}$ was located at 724.69 eV, and the peak of Fe $2p_{3/2}$ was composed of two splitting peaks at 711.38 and 714.05 eV, indicating that iron in ND/LDH appeared as $Fe^{3+}$ with the outermost electron configuration of $3d^5$ [5,43].

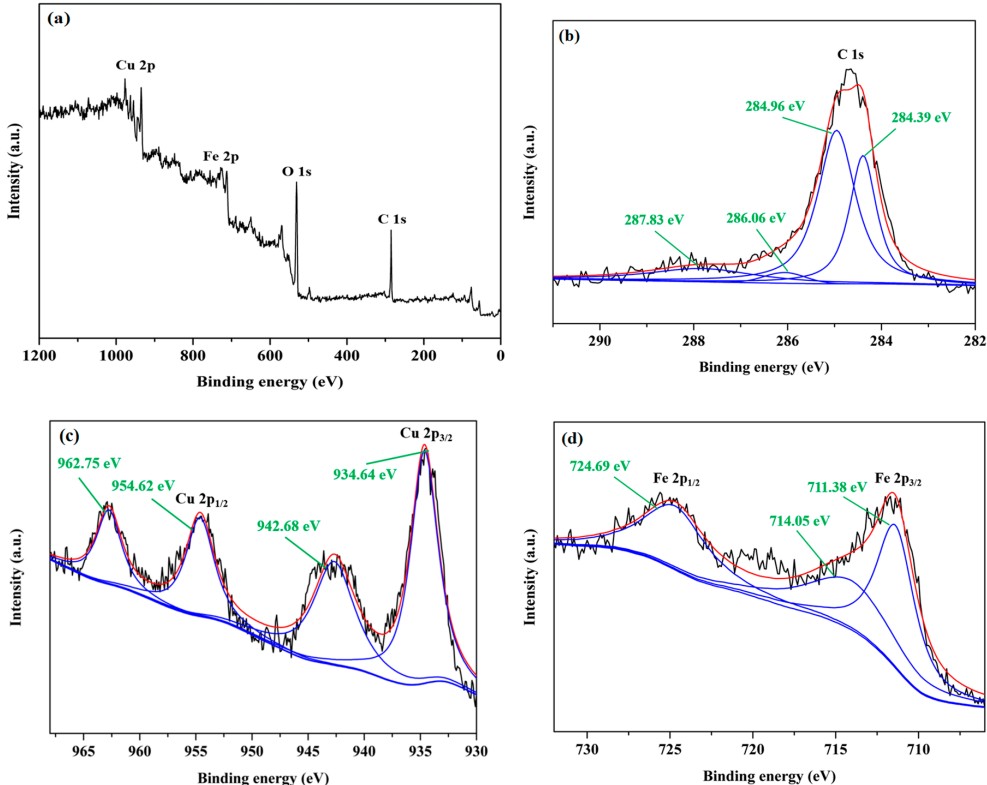

**Figure 3.** (**a**) Full-range X-ray photoelectron spectra (XPS) spectra of ND/LDH; (**b**) XPS peaks for C 1s; (**c**) XPS peaks for Cu 2p; (**d**) XPS peaks for Fe 2p.

FTIR spectra of LDH and ND/LDH were recorded using KBr pellets and the results are presented in Figure 4. The sharp peak at 3422 cm$^{-1}$ was ascribed to the stretching vibration of hydroxyl, which would come from the metal layer of LDH or ND [33]. A carbon dioxide absorption peak was also found at 2373 cm$^{-1}$. The peak at 1637 cm$^{-1}$ was ascribed to the bending vibration of hydroxyl which is located at the interlayer of LDH or ND [33]. The peak at 1385 cm$^{-1}$ was ascribed to the stretching vibration of the C-O structure which was from the carbonate located in the anionic interlayer of LDH, or from the oxygen-containing group on the ND surface. The peaks at 1050 and 505 cm$^{-1}$ were ascribed to Cu-O-Fe stretching vibration and metallic bond vibration (M-OH), respectively [1,37,44]. These results also indicated that the as-prepared ND/LDH possessed a hydrotalcite-like structure, and the copper and iron atoms in the metal layer were connected by an oxygen atom. Moreover, these results indicated that ND and LDH were hybridized successfully.

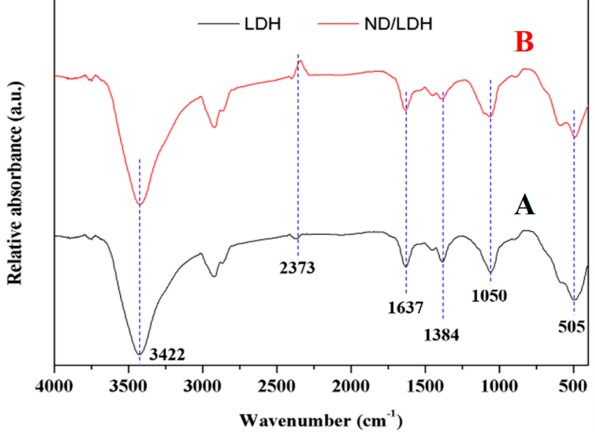

**Figure 4.** Fourier transform infrared (FTIR) spectra of LDH (**A**) and ND/LDH (**B**).

The surface area and pore structure of LDH and ND/LDH were investigated by the $N_2$ adsorption-desorption isotherm, respectively (Figure 5). Both LDH and ND/LDH exhibited typical IV isotherms with H3-type hysteresis loops ($P/P_0 > 0.4$), indicating the presence of mesopores [35]. Furthermore, no limiting adsorption at higher $P/P_0$ was observed, indicative of the existence of macropores [45]. The Brunauer-Emmett-Teller (BET) surface area of ND/LDH ($S_{BET}$ = 99.16 $m^2$/g) was clearly higher than that of LDH ($S_{BET}$ = 63.62 $m^2$/g), which would facilitate the photocatalytic activity of ND/LDH. Additionally, pore size distributions were determined using the Barrett-Joyner-Halenda (BJH) method. As is shown in Figure 5b,d, 2–20 nm was observed as the main pore size for both LDH and ND/LDH, indicating that the as-prepared ND/LDH belonged as a mesoporous material. Therefore, it can be inferred that ND/LDH, rather than bare LDH, would possess excellent catalytic activity due to its mesoporous structure and higher surface area.

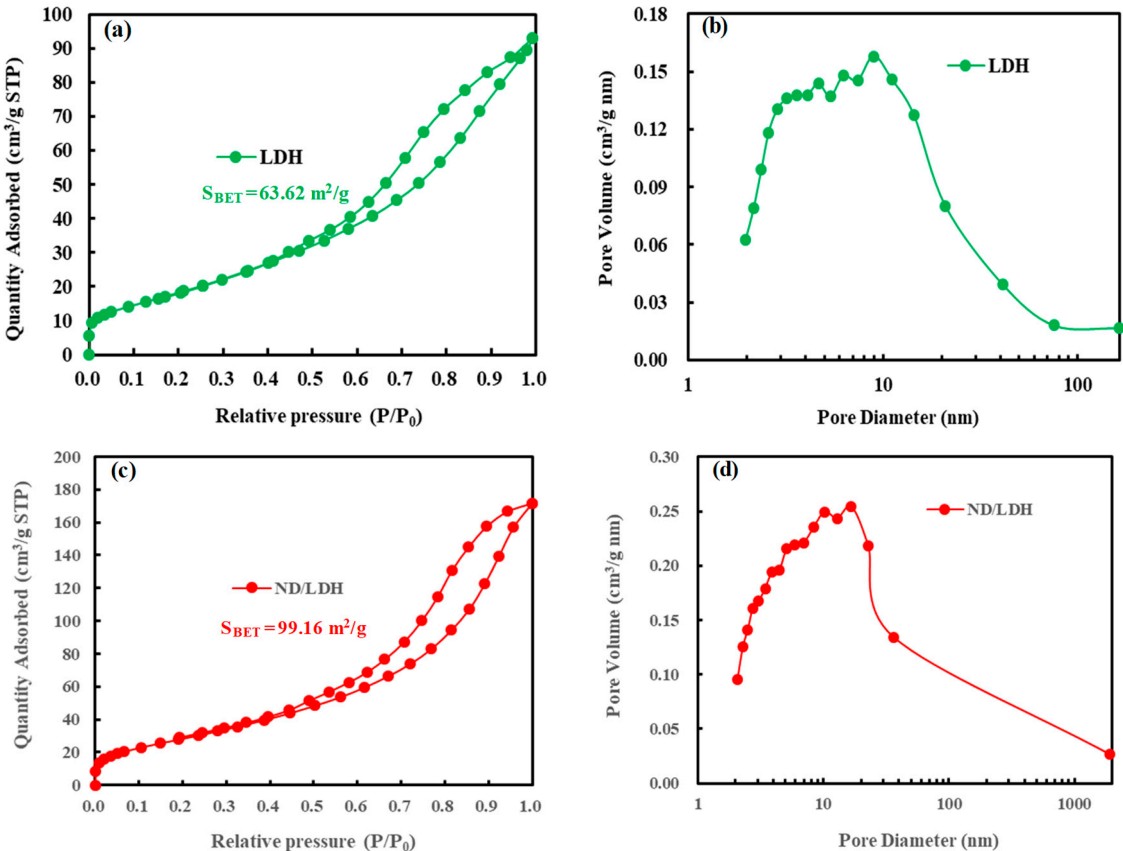

**Figure 5.** (**a**) Nitrogen adsorption-desorption isotherm of LDH; (**b**) Pore diameter distribution curve of LDH; (**c**) Nitrogen adsorption-desorption isotherm of ND/LDH; (**d**) Pore diameter distribution curve of ND/LDH.

## 2.2. Photocatalytic Activity of ND/LDH

In order to investigate the photocatalytic activities of ND/LDH under visible-light irradiation, methylene blue (MB) dye was employed as the targeted pollutant. As shown in Figure 6, in a dark environment (Figure 6a), both LDH and ND/LDH had no removal effectiveness on MB, indicating no absorption of MB on LDH or ND/LDH. Bare $H_2O_2$ had a little removal effectiveness on MB in darkness, which would be ascribed to its intrinsic oxidability. However, the assembly of LDH combined with $H_2O_2$ (LDH/$H_2O_2$) demonstrated high activity on MB removal in darkness and the removal rate of MB at 120 min reached 36.7%. Meanwhile, ND/LDH/$H_2O_2$ demonstrated slightly higher activity on MB removal in darkness and the removal rate of MB at 120 min reached 38.4%.

When the Fenton reaction systems were irradiated by visible-light (Figure 6c), both LDH and ND/LDH still had no removal effectiveness on MB, while bare $H_2O_2$ showed better MB removal than in darkness. To our excitement, both LDH/$H_2O_2$ and ND/LDH/$H_2O_2$ demonstrated dramatically higher activity on MB removal under visible-light irradiation than in darkness, and the removal rate of MB at 120 min reached 78.3% and 93.5%, respectively. The kinetics of MB degradation were also investigated. The experimental data were fitted with the pseudo-first-order kinetic equation as expressed by Equation (18). The results clearly showed that the apparent rate constant ($k$) of ND/LDH/$H_2O_2$ for MB degradation under visible-light irradiation ($23.3 \times 10^{-3}$ min$^{-1}$) was higher than that of LDH/$H_2O_2$ ($12.9 \times 10^{-3}$ min$^{-1}$) (Table 1). These results demonstrated that the hybridization of ND/LDH, rather than bare LDH, positively improved the efficiency of the visible-light driven photo-Fenton ($\lambda \geq 420$ nm).

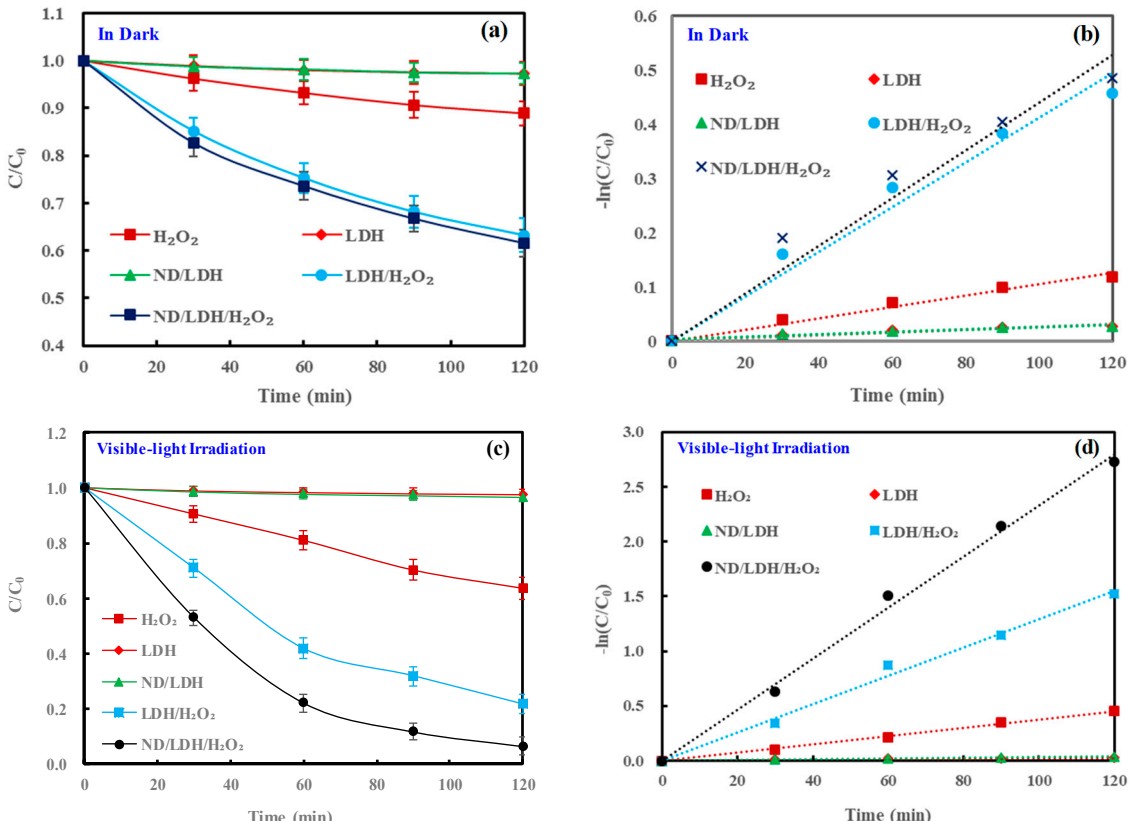

**Figure 6.** (**a**) Degradation of methylene blue (MB) in darkness; (**b**) pseudo-first-order kinetic fit for MB degradation in darkness; (**c**) photocatalytic degradation of MB under visible-light irradiation; and (**d**), pseudo-first-order kinetic fit for MB photodegradation under visible-light irradiation. (Error bars represent standard deviation of triplicate runs).

Many studies in the literature have reported that the concentration of a heterogeneous photocatalyst had a dramatic impact on photocatalytic efficiency [46]. Thus, the effects of ND/LDH dosage on MB removal under visible-light irradiation were investigated, with a concentration range of ND/LDH from 0.0027 to 0.2667 g/L. Clearly, 0.0667 g/L was the optimal concentration for ND/LDH to photocatalytically degrade MB dye (Figure 7), and the corresponding apparent rate constant ($k$) was the highest ($23.3 \times 10^{-3}$ min$^{-1}$) (Table 1). It should be noted that the turbidity of the reaction solution gradually increased, along with the increase in the ND/LDH dosage. Excessive high-turbidity can weaken the light penetrating into a photocatalytic solution, resulting in low photocatalytic efficiency [46].

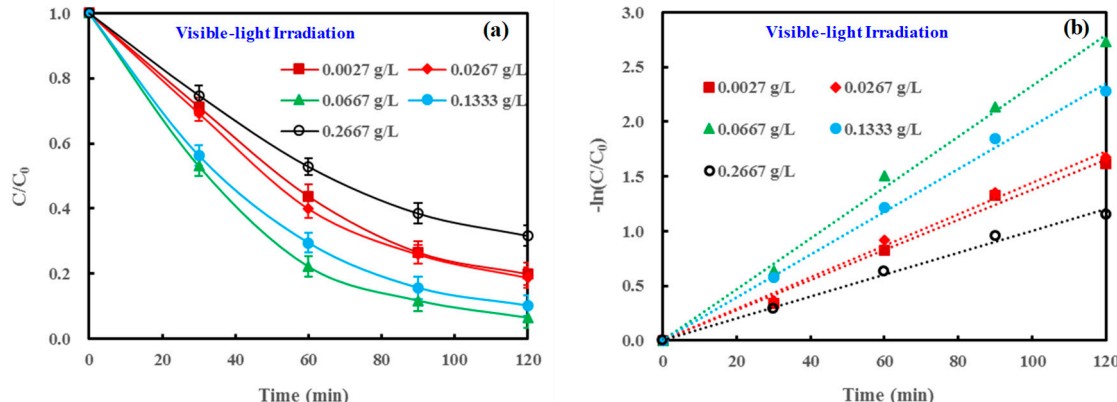

**Figure 7.** (**a**) Effect of ND/LDH dosage on MB removal under visible-light irradiation; and (**b**) pseudo-first-order kinetic fit for MB removal. (Error bars represent standard deviation of triplicate runs).

**Table 1.** Pseudo-first-order rate constants (*k*) of MB degradation and respective regression coefficients (*r*2) in different reaction systems.

| Photocatalytic Reagents | Dosage of LDH or ND/LDH (g/L) | Dosage of Radical Scavenger | Irradiation Condition | $k$ ($10^{-3}$ $\text{min}^{-1}$) | *r*2 |
|---|---|---|---|---|---|
| $H_2O_2$ | - | - | In dark | 0.1 | 0.9786 |
| $H_2O_2$ | - | - | VLI | 3.8 | 0.9955 |
| LDH | 0.0667 | - | In dark | 0.2 | 0.9388 |
| LDH | 0.0667 | - | VLI | 0.2 | 09012 |
| ND/LDH | 0.0667 | - | In dark | 0.3 | 0.9105 |
| ND/LDH | 0.0667 | - | VLI | 0.3 | 0.9199 |
| LDH/$H_2O_2$ | 0.0667 | - | In dark | 4.1 | 0.9685 |
| LDH/$H_2O_2$ | 0.0667 | - | VLI | 12.9 | 0.9918 |
| ND/LDH/$H_2O_2$ | 0.0667 | - | In dark | 4.4 | 0.9507 |
| ND/LDH/$H_2O_2$ | 0.0667 | - | VLI | 23.3 | 0.9955 |
| ND/LDH/$H_2O_2$ | 0.0027 | - | VLI | 13.8 | 0.9919 |
| ND/LDH/$H_2O_2$ | 0.0267 | - | VLI | 14.4 | 0.9931 |
| ND/LDH/$H_2O_2$ | 0.1333 | - | VLI | 19.6 | 0.9956 |
| ND/LDH/$H_2O_2$ | 0.2667 | - | VLI | 10.0 | 0.9924 |
| ND/LDH/$H_2O_2$ | 0.0667 | 0.5 mL/L of TBA | VLI | 19.3 | 0.9929 |
| ND/LDH/$H_2O_2$ | 0.0667 | 2.0 mL/L of TBA | VLI | 10.2 | 0.9850 |
| ND/LDH/$H_2O_2$ | 0.0667 | 5.0 mL/L of TBA | VLI | 3.5 | 0.9073 |
| ND/LDH/$H_2O_2$ | 0.0667 | 0.5 mL/L of EtOH | VLI | 12.3 | 0.9963 |
| ND/LDH/$H_2O_2$ | 0.0667 | 2.0 mL/L of EtOH | VLI | 5.5 | 0.9676 |
| ND/LDH/$H_2O_2$ | 0.0667 | 5.0 mL/L of EtOH | VLI | 1.0 | 0.9461 |

The dosages of $H_2O_2$ were all 19.6 mmol/L in each reaction; VLI means visible-light irradiation.

The stability of the photocatalyst is also a pivotal factor during the catalytic process. In order to investigate the stability of ND/LDH, the previous sample was reused in the same experimental conditions. Figure 8a represents the photocatalytic degradation cycle times of MB on ND/LDH. After five cycles, the degradation efficiency of MB on ND/LDH remained above 93.2%, indicating high stability and photocatalytic activity of ND/LDH over repeated use. Additionally, the heavy metals, copper and iron, leaching from ND/LDH were investigated during the first use process. Figure 8b shows that a little copper was leaching from ND/LDH, while no iron was detected throughout the whole photodegradation process. Although it has been reported that leakage of copper would reduce the redox cycle efficiency of $Cu^{3+}/Cu^{2+}$ [3,10], there was so little leakage of copper that this did not decrease the photocatalytic activity of ND/LDH.

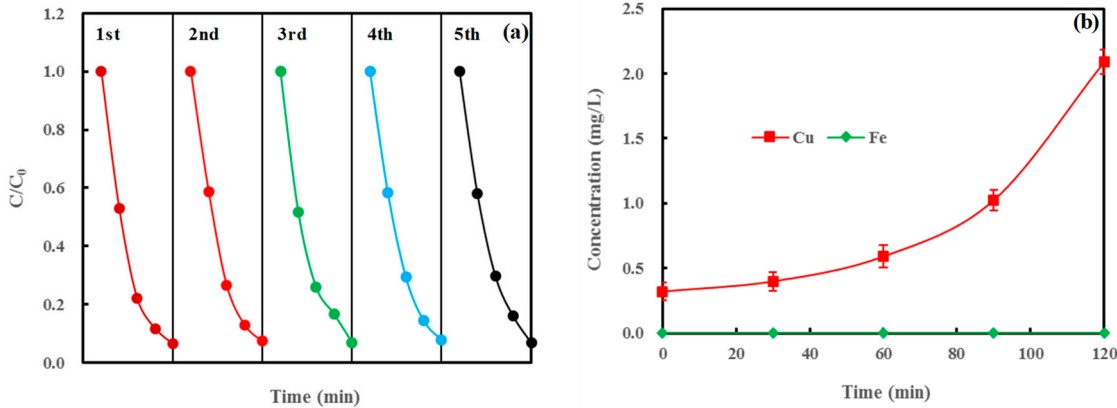

**Figure 8.** (**a**) Photocatalytic degradation cycle of MB on ND/LDH; and (**b**) leakage of Cu and Fe from ND/LDH in the reaction solution. (Error bars represent standard deviation of triplicate runs).

### 2.3. Photocatalytic Mechanism of ND/LDH

It is an indisputable fact that the ND/LDH/$H_2O_2$ irradiated by visible-light can enhance the photo-Fenton degradation of MB dye efficiently. To answer the question that there was specious decoloration or further mineralization for MB, the total organic carbon (TOC) concentrations of the MB solution were monitored dynamically during the photodegradation process. The results in Figure 9 show that the TOC of the MB solution treated with ND/LDH/$H_2O_2$ with visible-light irradiation decreased regularly along with the reaction time, and the removal rate of TOC at 120 min was 89.7%, a little lower than the removal rate of MB (93.5%) quantified by decoloring degree (Figure 6). For other reported photo-Fenton catalysts, such as FePcS-LDH and CdS-carbon nanotube/$TiO_2$, the TOC removals of MB at the reaction time of 120 min were only 50.0% and 84.0%, respectively [47,48]. These results confirmed that MB was positively mineralized in the ND/LDH/$H_2O_2$/visible-light system, although there was a little residue of TOC, and that ND/LDH was more efficient than previously reported photo-Fenton catalysts for TOC removal.

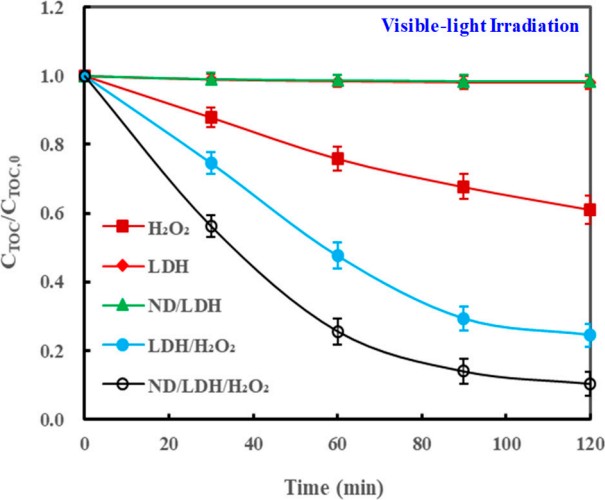

**Figure 9.** Variation of total organic carbon (TOC) of MB solution during the photodegradation process.

Generally, the mineralization of organic pollutants results from the oxidation of free radicals in AOPs. For the Fenton reaction, the hydroxyl radical is the main active-species for oxidizing the organic pollutant and leads to mineralization [6–8]. Hence, *t*-butanol (TBA) and absolute ethanol (EtOH) which are often used as hydroxyl radical scavengers, were employed to investigate the mechanism of ND/LDH/$H_2O_2$/visible-light treatment [49,50]. The results in Figure 10 clearly showed that both

TBA and EtOH can inhibit the degradation of MB, and the higher concentration of the scavengers was, the stronger the inhibition. Correspondingly, the apparent rate constant (*k*) of MB degradation also declined dramatically with the increasing addition of hydroxyl radical scavengers (Table 1). Therefore, it was verified that the hydroxyl radical was the main active-species for MB degradation in the ND/LDH/H$_2$O$_2$/visible-light system.

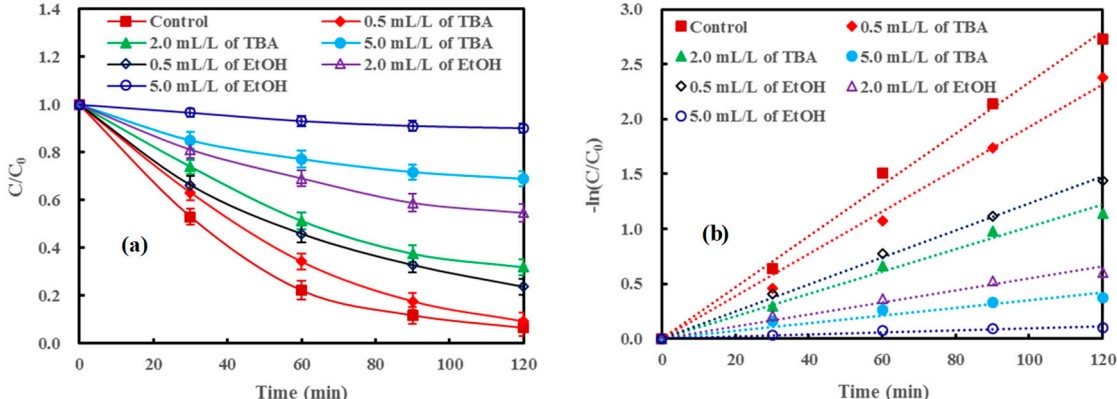

**Figure 10.** (**a**) Effect of free radical scavengers (*t*-butanol, TBA; absolute ethanol, EtOH) on MB removal under visible-light irradiation; and (**b**) pseudo-first-order kinetic fit for MB removal. (Error bars represent standard deviation of triplicate runs).

The results of XPS indicated that bivalent copper and trivalent iron were the main elementary forms in ND/LDH (Figure 3). As is commonly known, the redox cycle of Fe$^{3+}$/Fe$^{2+}$ (E(Fe$^{3+}$/Fe$^{2+}$) = 0.77 eV vs. NHE) is the fundamental driver for the traditional Fenton reaction [3]. It also had been reported that the redox cycle of Cu$^{3+}$/Cu$^{2+}$ (E(Cu$^{3+}$/Cu$^{2+}$) = 2.30 eV vs. NHE) rather than Cu$^{2+}$/Cu$^+$ (E(Cu$^{2+}$/Cu$^+$) = 0.15 eV vs. NHE) would more efficiently catalyze H$_2$O$_2$ to produce hydroxyl radicals, because such a reaction is thermodynamically feasible for activating H$_2$O$_2$ [6,10,21,22,51]. Consequently, the redox cycles of Cu$^{3+}$/Cu$^{2+}$, Cu$^{2+}$/Cu$^+$ and Fe$^{3+}$/Fe$^{2+}$ in ND/LDH would synergistically catalyze H$_2$O$_2$ to produce hydroxyl radicals for MB degradation. The detailed reaction mechanism was described as follows [3,36,52]:

$$Cu^{2+} + H_2O_2 \rightarrow Cu^{3+} + HO\cdot + OH^- \tag{1}$$

$$Fe^{3+} + H_2O_2 \rightarrow HO_2\cdot + Fe^{2+} + H^+ \tag{2}$$

$$Cu^{3+} + H_2O_2 \rightarrow Cu^{2+} + HO_2\cdot + H^+ \tag{3}$$

$$HO_2\cdot + H_2O_2 \rightarrow O_2 + H_2O + HO \tag{4}$$

$$Cu^{3+} + Fe^{2+} \rightarrow Cu^{2+} + Fe^{3+} \tag{5}$$

$$Fe^{2+} + H_2O_2 \rightarrow Fe^{3+} + HO\cdot + OH^- \tag{6}$$

$$Cu^{2+} + H_2O_2 \rightarrow Cu^+ + HO_2\cdot + H^+ \tag{7}$$

$$Cu^+ + H_2O_2 \rightarrow Cu^{2+} + HO\cdot + HO^- \tag{8}$$

$$HO\cdot + MB\ dye \rightarrow Degradation \tag{9}$$

When the Fenton reaction performed by ND/LDH/H$_2$O$_2$ was irradiated by visible-light ($\lambda \geq$ 420 nm), the removal rate of MB dramatically increased in the photo-Fenton reaction system (Figure 6). To investigate the photocatalytic mechanism, the optical properties of LDH and ND/LDH were determined by UV-Vis diffuse reflectance spectra (DRS) from 190 to 800 nm. The results showed four strong absorption bands in the spectral regions, of 190–250 nm, 280–400 nm, 450–680 nm and

770–800 nm, which can be attributed to the ligand-to-metal charge transfer (LMCT) that arises from the 2p orbital of oxygen to the 3d orbitals of Cu/Fe (Figure 11a) [53–56]. Owing to the orbital split caused by crystal electric field action, the absorption band located at 190–250 nm should be ascribed to the electron transfer from σ orbital of O 2p to σ* (e$_g$) orbital of Cu/Fe 3d, the band located at 280–400 nm ascribed to the electron transfer from σ orbital of O 2p to π* (t$_{2g}$) orbital of Cu/Fe 3d, the band located at 450–680 nm ascribed to the electron transfer from π orbital of O 2p to σ* (e$_g$) orbital or π* (t$_{2g}$) orbital of Cu/Fe 3d, and the band located at 770–800 nm ascribed to the electron transfer from π* (t$_{2g}$) orbital to σ* (e$_g$) orbital of Cu/Fe 3d (d-d internal electron transition) [16]. These results demonstrated that both LDH and ND/LDH can absorb visible-light efficiently, suggesting small band gaps of LDH and ND/LDH, which were associated with the excellent photocatalytic performance on MB removal under visible-light irradiation. However, the absorbance curve of LDH declined consistently after 675 nm, while the absorbance curve of ND/LDH began to decline at 770 nm, suggesting that the band gap energy of ND/LDH (E$_g$(ND/LDH)) was smaller than the band gap energy of LDH (E$_g$(LDH)). Generally, the band gap energy of a crystalline semiconductor can be calculated from the following equation [16,57,58]:

$$(\alpha \times h \times \upsilon)^{1/n} = A \times (h \times \upsilon - Eg) \tag{10}$$

where α, h, υ, E$_g$ and A are the the absorption coefficient, Planck's constant, light frequency, band gap energy and a constant, respectively. Aside from these, n is determined by the type of optical transition of a semiconductor (n = 1/2 for direct transition and n = 2 for indirect transition) [59]. LDHs have previously been found to have direct transitions [17]. Therefore, the plots of $(\alpha \times h \times \upsilon)^2$ versus h × υ (Kubelka-Munk function as a function of light energy) gave the values of the band gap energy by extrapolating the straight line to the h × υ axis intercept, as shown in Figure 11b. The estimated E$_g$(LDH) was found to be 1.06 eV, suggesting a photoelectron transfer from Cu/Fe 3d t$_{2g}$ to Cu/Fe 3d e$_g$. However, the estimated E$_g$(ND/LDH) was found to be 0.94 eV, clearly lower than E$_g$(LDH), which should arise from the hybridization of ND and LDH.

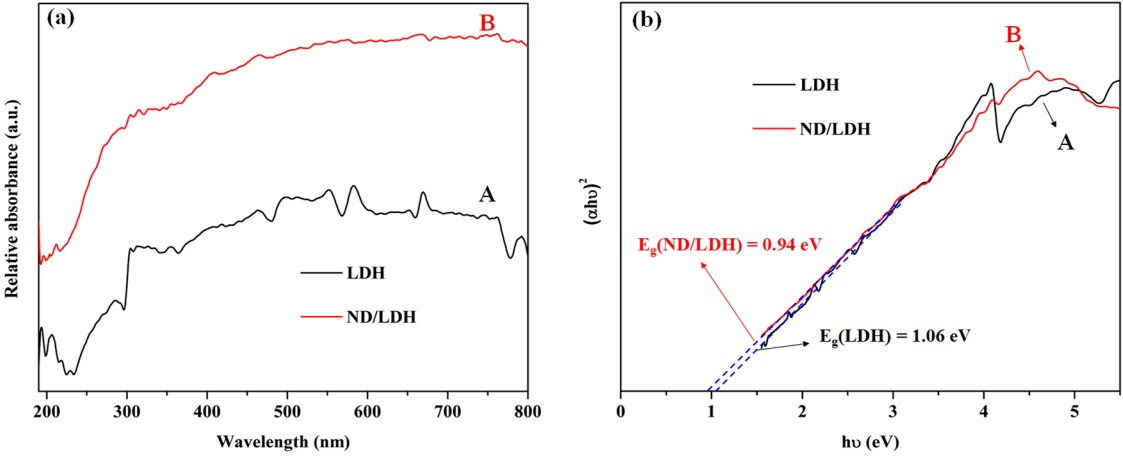

**Figure 11.** UV-Vis diffuse reflectance spectra (DRS) (**a**) and calculated the band gap (E$_g$) (**b**) of LDH (**A**) and ND/LDH (**B**).

Since E$_g$ determines the maximum absorption wavelength of a semiconductor photocatalyst, the reducing capacity of the photogenerated electron and the oxidizing capacity of photogenerated hole are decided by the positions of conduction band (CB) and valence band (VB), respectively. In this study, the VB of LDH and ND/LDH was measured using XPS valence spectra (Figure 12) with the empirical equation from [60,61]. The VB maximum positions (E$_{VB}$) of both LDH and ND/LDH were determined to all be 1.48 eV (vs. NHE). Subsequently, the CB minimum position (E$_{CB}$) could be estimated by the following equation [37,62]:

$$E_{CB} = E_{VB} - Eg \tag{11}$$

Then, the $E_{CB}$ of LDH and $E_{CB}$ of ND/LDH were calculated to be 0.42 and 0.54 eV (vs. NHE), respectively.

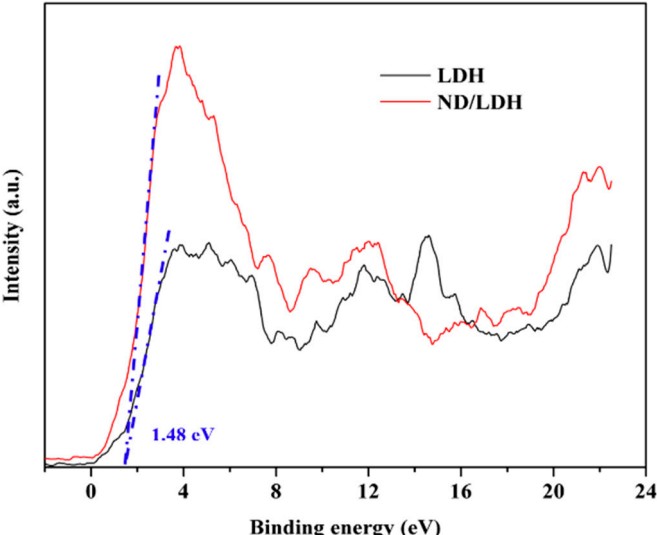

**Figure 12.** Valence band (VB) XPS spectra of LDH and ND/LDH.

It was known that the band gap energy of bulk diamond is ~5.4 eV, and only deep UV-light can excite the valence electrons from VB to CB [58]. Moreover, because the electron diffusion lengths in a polycrystalline diamond are only on the order of 1 μm, many of the impinging photons are absorbed too deeply in the bulk for the resulting CB electrons to reach the surface before recombination [63]. Therefore, it is hard to utilize bulk diamond for photocatalysis directly. Fortunately, ND has a smaller particle size (typically 4–5 nm) and higher surface area (BET surface areas of around 300 m$^2$/g) [23–26] and exhibits a lower band gap due to the graphitized structure and the oxygen-containing groups on the surface of ND [23]. As is commonly known, the bound π electron in sp2-hybridized carbon atoms is weak and forms a free electron when π orbitals overlap. In this situation, the graphitized carbon atoms are bound to influence the electronic structure of ND. It has been reported that the band gap energy of ND with a fraction of sp2-hybridized carbon can be calculated by Equation (12) [64]:

$$\frac{1}{E_g(x)} = \frac{1-x}{E_g(0)} + \frac{x}{E_g(1)} \tag{12}$$

where x is the ratio of the sp2 carbon content to the amount of sp3 and sp2 carbon atoms. $E_g(x)$ is the corresponding band gap energy, with $E_g(0)$ = 5.4 eV (diamond) and $E_g(1)$ = 1.4 eV (graphite) [65]. It has been reported that the x value of detonation ND is ~0.29 [23]. Therefore, the calculated band gap energy of ND was 2.95 eV. It is also known that oxygen-containing groups with substantial electron density and electron-donating ability on the surface of ND will enlarge its electron affinity and lower its CB. Thus, the band gap of ND would narrow further, and has been reported to be ~2.28 eV [23]. Moreover, ND belongs to the n-type semiconductors so that the Fermi level ($E_F$) was more closed to CB [23].

When the bare LDH was irradiated by visible-light, the ground state electron at VB (Cu/Fe 3d $t_{2g}$) was excited and leapt onto CB (Cu/Fe 3d $e_g$) (Figure 13). Because the electric potential of $E_{VB}$ (1.48 eV vs. NHE) was lower than the electric potential of E(OH$^-$/HO·) (1.99 eV vs. NHE) [66,67], the photogenerated hole was unable to oxidize the OH$^-$/H$_2$O to produce strong oxidizing hydroxyl radical, consistent with the experimental fact that bare LDH cannot degrade MB dye under visible-light irradiation (Figure 6). However, the electric potential of $E_{CB}$ (0.42 eV vs. NHE) was lower than the electric potentials of both E(Fe$^{3+}$/Fe$^{2+}$) (0.77 eV vs. NHE) and E(Cu$^{3+}$/Cu$^{2+}$) (2.30 eV vs. NHE), so the photoelectrons at CB, rather than at VB, could more easily reduce Cu$^{3+}$/Fe$^{3+}$ into Cu$^{2+}$/Fe$^{2+}$ which

were the key active-species for catalyzing $H_2O_2$-generating hydroxyl radicals for oxidative degradation of MB dye [3,68,69]. Therefore, the essence of visible-light driven photo-Fenton degradation of MB on bare LDH was that the excited state photoelectron can efficiently accelerate the redox cycles of $Cu^{3+}/Cu^{2+}$ and $Fe^{3+}/Fe^{2+}$. In addition, according to the metal-ligand charge transfer (MLCT) mechanism, $Fe^{3+}$, $Cu^{3+}$ or $Cu^{2+}$ which is combined with hydroxyls can be photoreduced under UV-Vis light irradiation and generate hydroxyl radicals via the following reactions [47,48,70,71]:

$$Fe(OH)^{2+} + h \times \upsilon \rightarrow Fe^{2+} + HO\cdot \tag{13}$$

$$Cu(OH)^{2+} + h \times \upsilon \rightarrow Cu^{2+} + HO\cdot \tag{14}$$

$$Cu(OH)^{+} + h \times \upsilon \rightarrow Cu^{+} + HO\cdot \tag{15}$$

Since the ND and LDH were hybridized as a novel heterojunction with a common $E_F$, the photoelectrons at the CB of LDH ($CB_{LDH}$) could spontaneously migrate onto the low-energy CB of ND ($CB_{ND}$), promoting the effective separation photo-induced charges. This is because ND can quickly capture the photo-generated carriers due to its rather small Bohr radius and large exciton binding energy [23]. Thus, the photoelectrons at $CB_{ND}$ had sufficient time to enhance the redox cycles of $Cu^{3+}/Cu^{2+}$ and $Fe^{3+}/Fe^{2+}$, further improving the photo-Fenton efficiency. Beyond that, ND can also be excited by visible-light and the photoelectron can also enhance the redox cycles of $Cu^{3+}/Cu^{2+}$ and $Fe^{3+}/Fe^{2+}$. Moreover, the photogenerated holes can oxidize $H_2O_2$ to produce hydroxyl radicals which was the key active-species for MB mineralization. Of course, it should be noted that dye photosensitization may be another way to degrade MB.

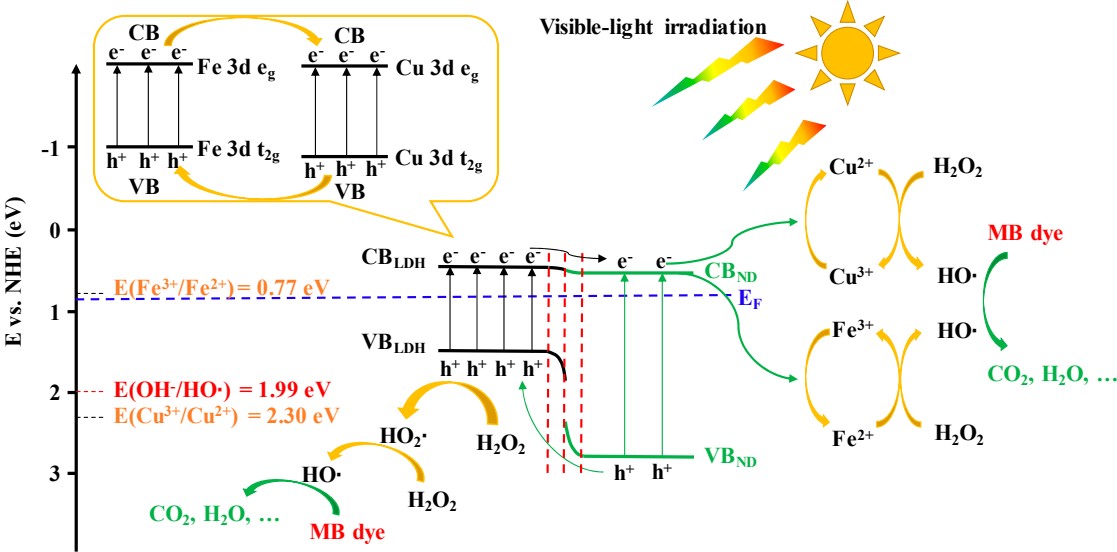

**Figure 13.** Proposed mechanism of the charge transfer in ND/LDH for the photo-Fenton degradation of MB dye under visible-light irradiation.

## 3. Materials and methods

### 3.1. Chemicals

$CuSO_4$ (99.0%), $FeCl_3\cdot \times 6H_2O$ (99.0%), NaOH (99.8%), methylene blue (MB) (99.0%), $H_2O_2$ (30.0%), absolute ethanol (EtOH) (99.7%) and *t*-butanol (TBA) (98%) were purchased locally from Sinopharm Chemical Reagent Co., Ltd. (Shangai, China) and were used as received without further purification. ND (detonation nanodiamond) purchased from Tianjin Qianyu super hard technology Co., Ltd. (Tianjin, China) was used in this study and the mean particle size was ~5 nm.

### 3.2. Synthesis of Powdered ND/LDH

The powdered ND/LDH was prepared using a simple coprecipitation method. In a general procedure, 400 mL metal salt solution A, containing 0.500 g $CuSO_4$ and 0.423 g $FeCl_3 \cdot \times 6H_2O$ (Cu/Fe molar ratio was 2:1), and 200 mL NaOH solution B containing 0.0028 g ND (pH = 9), were first prepared. Then, solution A was added dropwise into solution B with continuous stirring at 60 °C. It should be noted that the pH of solution B was controlled at 9 through adding concentrated NaOH solution (1.0 M) from beginning to end. After that, the ~600 mL dark-brown suspension liquid was evaporated to ~60 mL with continuous stirring at 60 °C. Subsequently, the concentrated ~60 mL dark-brown suspension was transferred into a 100 mL Teflon lined stainless steel laboratory autoclave and heated in a furnace with a constant temperature of 100 °C for a period of 10 h. After cooling to room temperature, the dark-brown product was obtained by centrifugation at 10,000 rpm for 10 min. Finally, this dark-brown product was dried at 80 °C overnight, ground into fine powder with an agate mortar, and named ND/LDH. In addition, bare LDH was also synthesized as the experimental control according to above synthesis procedures, except that the NaOH solution B did not contain ND.

### 3.3. Characterization of ND/LDH

The crystal lattice structure was shown by XRD (XRD-6000 Shimadzu, Kyoto, Japan) and HRTEM (JEOL JEM2100, Tokyo, Japan). The surface elemental chemical state was characterized by XPS (ESCALAB250, Thermo Fisher Scientific, Massachusetts, USA) using monochromated Al K$\alpha$ X-ray (1486.6 eV) as the excitation source radiation. Fourier transform infrared (FTIR) spectroscopy was conducted using the Nicolet iN 10 (Thermo Fisher) spectrometer with the resolution of 4 cm$^{-1}$. The $N_2$ adsorption-desorption isotherm was measured in a micromeritics JW-BK132F analyzer (Beijing JWGB Sci. & Tech. Co., Ltd., Beijing, China) at −196 °C. Brunauer-Emmett-Teller (BET) specific surface area, pore volume and pore diameter were evaluated from the adsorption branch of the nitrogen isotherm based on the BET and Barrett-Joyner-Halenda (BJH) model. UV-Vis diffuse reflectance spectra (DRS) were collected on a Shimadzu spectrophotometer (UV-2550), and $BaSO_4$ was used as a reference.

### 3.4. Photocatalytic Degradation Experiments

The photocatalytic activity of ND/LDH was evaluated with MB. A 500 W Xe lamp (GXZ500, Shanghai Jiguang special lighting electric appliance factory, Shanghai, China) with an UV filter (420 nm) was used as the simulated visible-light source. The distance between the lamp and the suspension was 15 cm. Typically, 5.0 mg of ND/LDH and 150 μL of $H_2O_2$ were simultaneously added into an aqueous MB solution (75 mL, 10 mg/L) without initial pH adjustment, then illuminated under visible-light for MB photocatalytic degradation. At a certain time interval (30 min), a 3 mL suspension was withdrawn from the reaction and subjected to centrifugation. The MB concentration was monitored using a UV-Vis spectrophotometer (UV-1240 Shimadzu) at the wavelength of 664 nm. In addition, catalyst dosage was optimized, catalyst stability was investigated based on catalyst reuse and metallic leaching, and total organic carbon (TOC, mg/L) of the MB solution was determined by a TOC analyzer (SSM-5000A Shimadzu) during the photocatalytic degradation process. To reveal the photocatalytic mechanism, typical radical scavengers such as EtOH and TBA were added into the photo-Fenton reactions to verify the active-species.

The degradation efficiency of MB can be defined as follows [72,73]:

$$\text{Degradation}_{(\%)} = (1 - C/C_0) \times 100\% \tag{16}$$

where C and $C_0$ represent the MB concentration at time t and the initial concentration of MB dye, respectively. The error analysis was calculated using the following standard deviation formula [74]:

$$\sigma^2 = \lim_{n \to \infty} \left[ \frac{1}{N} \sum (x_i - \mu)^2 \right] \tag{17}$$

where σ is the standard deviation, $x_i$ is the observed value and μ is the mean value. The kinetics of degradation of MB were also investigated. The experimental data were fitted with the pseudo-first-order kinetic equation as expressed by Equation (18) [73,74]:

$$\ln(C/C_0) = -k \times t \tag{18}$$

where *k* is the apparent reaction rate constant ($min^{-1}$).

## 4. Conclusions

In this study, a simple coprecipitation method was adopted to synthesize a ND/LDH catalyst for the photo-Fenton reaction. The ND/LDH possessed a hydrotalcite-like structure, larger specific surface area and strong absorption of visible-light. The removal rate of MB in the ND/LDH/$H_2O_2$ system under visible-light irradiation was dramatically higher than that in the LDH/$H_2O_2$/visible-light system. Moreover, the ND/LDH maintained good stability and photocatalytic capacity after using five times. For bare LDH, the $E_{VB}$ was lower than $E(OH^-/HO·)$ so the photogenerated hole was unable to oxidize $H_2O/OH^-$ to produce the hydroxyl radical, whereas the photogenerated electron could efficiently accelerate the redox cycles of $Cu^{3+}/Cu^{2+}$ and $Fe^{3+}/Fe^{2+}$ to promote the decomposition of $H_2O_2$ into hydroxyl radicals. In the heterojunction of ND/LDH, the spontaneous migration of photoelectrons from the $CB_{LDH}$ (Cu/Fe 3d $e_g$) to $CB_{ND}$ can effectively promote the separation of photo-induced charges, resulting in excellent photocatalytic activity on MB degradation. In addition, the photogenerated electrons of ND and the photogenerated holes can also react with $H_2O_2$ directly, or indirectly, to produce hydroxyl radicals, contributed to enhancing the photo-Fenton activity.

**Author Contributions:** Conceptualization, Y.A. and H.L.; methodology, L.L.; investigation and validation, L.L., S.L. and H.W.; catalysts characterization, X.S., J.L. and X.C.; mechanism analysis, L.L. and Y.A.; writing—original draft, L.L.

**Funding:** This study was supported by National Natural Science Foundations of China (41602249), Science and Technology Development Project of Jilin Province (20160520076JH), and College Students Innovation and Entrepreneurship Training Project of Jilin University (2017A64279).

**Conflicts of Interest:** The authors declare no conflict of interest.

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
