# Peer review of "Hybridization of Nanodiamond and CuFe-LDH as Heterogeneous Photoactivator for Visible-Light Driven Photo-Fenton Reaction: Photocatalytic Activity and Mechanism"

_catalysts, doi:10.3390/catal9020118_

Reviewer 1 Report

In general the presented data have been interesting, however authors should convince, that the proposed system for MB mineralisation has more advantages than the other used.

Above there are some comments or questions for the authors:

What is a role of ND in this hybrid system? Could you prove, that ND improve separation of free charges or enhance generation of OH radicals?

Could you show the plots of Kubelka-Munka function?

Some of the other Fenton reactions are missing. At the presence of H2O2 Fe2+ can be oxidised to Fe3+ and Cu+ to Cu2+ with formation of OH radicals. There was also no mentioned that Fe3+ or Cu2+ can be photoreduced during visible or UV light irradiation.

There is no reference to any literature about photo-Fenton process used for dyes decolorisation in order to compare the obtained results with the others, please refer to some papers such as ex.:

1)      X.Tang, Y. Liu, Heterogeneous photo-Fenton degradation of methylene blue under visible irradiation by iron tetrasulphophthalocyanine immobilized layered double hydroxide at circumneutral pH, Dyes and Pigments 134 (2016) 397-408.

2)      J. R. Kim, E. Kan, Heterogeneous photo-Fenton oxidation of methylene blue using CdS-carbon nanotube/TiO2 under visible light, Journal of Industrial and Engineering Chemistry 21 (2015) 644-652.

3)      B. Tryba, M. Piszcz, B. Grzmil, A. Pattek-Janczyk, A. W. Morawski, Photodecomposition of dyes on Fe-C-TiO2 photocatalysts under UV radiation supported by photo-Fenton process, J. Hazard. Mater., 162 (2009) 111-119.

4)      B. Tryba, A. W. Morawski, M. Inagaki, M. Toyoda, Mechanism of phenol decomposition on Fe-C-TiO2 and Fe-TiO2 photocatalysts via photo-Fenton process, J. Photochem. Photobiol. A: Chem. 179, (2006) 224-228.

The data should be discussed in the aspect of the other results obtained by different systems in order to show the priority of the proposed ND/LDH hybrid composite.

Author Response

Dear reviewer 1,

Thank you very much for reviewing our manuscript, and giving us so good questions and advice which I think actually can improve the quality of our paper. I am also particularly grateful to you for your acknowledgement on the interesting of our paper. Now, I have revised our manuscript according to your comments, all the revisions are highlighted in red, and the detailed revision processes are described as follows:

Comment 1:

In general the presented data have been interesting, however authors should convince, that the proposed system for MB mineralisation has more advantages than the other used.

Response to reviewer 1:

Thank you very much for your advice. According to the recommended literatures, I find that our photo-Fenton catalyst ND/LDH was more efficient than reported photo-Fenton catalysts on TOC removal, such as FePcS-LDH and CdS-carbon nanotube/TiO2. So, I have discussed this information in the part “2.3. Photocatalytic mechanism of ND/LDH” to highlight the advantage of ND/LDH.

Above there are some comments or questions for the authors:

Comment 2:

What is a role of ND in this hybrid system? Could you prove, that ND improve separation of free charges or enhance generation of OH radicals?

Response to reviewer 1:

Thank you very much for these two questions. ND was used to form heterojunction with LDH. As is known to all, heterojunction can facilitate the separation of photo-induced charges, resulting in high photocatalytic efficiency. It had been reported that the hybridization of ND with TiO2 can improve the separation of photo-induced charges and enhance the generation of hydroxyl radicals, the hybridization of ND with Cu2O also can improve the separation of photo-induced charges and enhance the photocatalytic hydrogen evolution. Therefore, we inferred that hybridization of ND with LDH would facilitate the photo-Fenton reaction due to the heterojunction effect. Fortunately, we found that ND/LDH was more efficient than LDH to catalyze photo-Fenton reaction. The HRTEM photo (Figure 1) showed that ND was hybridized with LDH successfully. According to the UV-vis DRS and VB XPS spectra, we proposed the charge transfer mechanism in ND/LDH. I think all the experimental results can better explain the charge transfer mechanism. Of course, if use Steady-state surface photovoltage spectroscopy, Transient-state surface photovoltage responses and Transient-state photoluminescence spectra to prove the separation of free charges, it will be much better. Therefore, we intend to use photovoltage and PL techniques to investigate the separation of free charges in ND heterojunction in future study.

Comment 3:

Could you show the plots of Kubelka-Munka function?

Response to reviewer 1:

Thank you very much for this comment. Kubelka-Munka function is used to describe the diffuse reflection law. The formula of Kubelka-Munka function is:

F(R)=(1- R)2/2 R∞    (1)

where F(R) is so-called Kubelka-Munka function, R= Rsample / Rstandard. For a sample, every monochromatic light is corresponding to a R value. Therefore, the plot of Kubelka-Munka function can be expressed by F(R) vs. wavelength (λ).

According to our experimental results, the plots of Kubelka-Munka function for LDH and ND/LDH are shown as follow:

Figure S1 The plots of Kubelka-Munka function for LDH and ND/LDH

Comparing Figure S1 and Figure 11 (a) in manuscript, we can find that they are equal, proving that Kubelka-Munka function is used to describe the absorbance spectra of UV-vis DRS.

Comment 3:

Some of the other Fenton reactions are missing. At the presence of H2O2 Fe2+ can be oxidised to Fe3+ and Cu+ to Cu2+ with formation of OH radicals. There was also no mentioned that Fe3+ or Cu2+ can be photoreduced during visible or UV light irradiation.

Response to reviewer 1:

Thank you very much for this comment. In general, redox pair Cu3+/Cu2+ [E(Cu3+/Cu2+) = 2.30 eV] is more efficient than Cu2+/Cu+ [E(Cu2+/Cu+) = 0.15 eV] to activate H2O2. Therefore, we just described the Cu3+/Cu2+-catalyzed Fenton reaction in the manuscript. However, it is a fact that Cu2+/Cu+ also can activate H2O2 to generate OH radicals. So, the Fenton reactions catalyzed by Cu2+/Cu+ are supplemented in the revised manuscript. In addition, the photoreduction of Fe3+, Cu3+ or Cu2+ is also discussed in the revised manuscript.

Comment 3:

There is no reference to any literature about photo-Fenton process used for dyes decolorisation in order to compare the obtained results with the others, please refer to some papers such as ex.

1)      X.Tang, Y. Liu, Heterogeneous photo-Fenton degradation of methylene blue under visible irradiation by iron tetrasulphophthalocyanine immobilized layered double hydroxide at circumneutral pH, Dyes and Pigments 134 (2016) 397-408.

2)      J. R. Kim, E. Kan, Heterogeneous photo-Fenton oxidation of methylene blue using CdS-carbon nanotube/TiO2 under visible light, Journal of Industrial and Engineering Chemistry 21 (2015) 644-652.

3)      B. Tryba, M. Piszcz, B. Grzmil, A. Pattek-Janczyk, A. W. Morawski, Photodecomposition of dyes on Fe-C-TiO2 photocatalysts under UV radiation supported by photo-Fenton process, J. Hazard. Mater., 162 (2009) 111-119.

4)      B. Tryba, A. W. Morawski, M. Inagaki, M. Toyoda, Mechanism of phenol decomposition on Fe-C-TiO2 and Fe-TiO2 photocatalysts via photo-Fenton process, J. Photochem. Photobiol. A: Chem. 179, (2006) 224-228.

The data should be discussed in the aspect of the other results obtained by different systems in order to show the priority of the proposed ND/LDH hybrid composite.

Response to reviewer 1:

Thank you very much for recommending literatures. I have carefully read these four papers and studied more photo-Fenton knowledge. The TOC removal efficiency in our study is compared with other results [such as in above literature 1) and 2)] in the revised manuscript.. And above four literatures are all cited in the revised manuscript.

Thank you very much again for your comments. I hope this revision is satisfied to you and our paper could be accepted soon.

Reviewer 2 Report

Review for Catalyst 424786

“Hybridization of Nanodiamond and CuFe-LDH as Heterogeneous Photoactivator for Visible-Light Driven Photo-Fenton Reaction: Photocatalytic Activity and Mechanism”

ARGUMENT: The objective of the paper " Hybridization of Nanodiamond and CuFe-LDH as Heterogeneous Photoactivator for Visible-Light Driven Photo-Fenton Reaction: Photocatalytic Activity and Mechanism" was to study a novel heterojunction nanodiamonds ND and CuFeLDH as heterogeneous photo-catalyst. The material structure was characterized by HRTEM, XRD, XPS, UV-vis and DRS. The photocatalytic performance of ND/LDH in the photo-Fenton catalytic reaction for MB degradation and the reaction mechanism were investigated, finding that the rate of removal of MB was effectively increased by the heterojunction ND/LDH.

QUALITY: The use of ND hybridized with other semiconductor photo-catalysts has a certain interest and novelty. The study is correctly designed and the analyses and the experimental are described with sufficient detail. The results are appropriately interpreted and the conclusions are supported by the results.

The data are clearly exposed, but authors may improve the clarity of paper modifying figures and tables basing on the following minor revisions.

ENGLISH: some misprints are present. A in-depth revision of the language is needed.

Minor revisions:

1) CuFe-LHD is defined also as the simpler LHD in the junction ND/LHD, or in the figures; whereas in the text and in the table1 was reported as CuFe-LHD or LDH. This can generate confusion. I suggest to always use the same form in the test, figures and table: always LHD or always CuFe-LHD.

2) line 85-87 “The crystal phase and crystallinity of ND, CuFe-LDH and ND/LDH were characterized by using a Rigaku X-ray diffractometer with Cu Kα radiation over a range of 2θ angles from 20° to 90° (Figure 2), respectively”

In experimental is reported a Shimatzu instrument. Is that a different XRD instrument? It should clarified and that information should be moved in the experimental.

3) line 127-129 “The peak at cm-1 was ascribed to the stretching vibration of C-O structure which was subject to carbonate in the anionic interlayer of CuFe- LDH or oxygen-containing group on ND surface”

This sentence is not clear at all. Authors should better write and better explain.

4) Fig 1 (A,B,C,D). letters ABCD in blue color are not visible in black and white printed version. In addition, they are to much small: they have to be increased in dimensions. The lattice fringes and the numbers in red color are not visible in black and white printed version. I suggest to change color in order to make the text legible even in the printed version.

5) Fig. 2  I suggest to number the three XRD spectra in order to recognize even in the printed version in black and white, and add the corresponding information in the caption.

6) Fig. 4  I suggest to number the two FTIR spectra in order to recognize even in the printed version in black and white, and add the corresponding information in the caption.

7) Table 1. The table is not clearly readable due to much text in the cells. I suggest:

a) eliminate the column 3 that reports the same information for all sample( move info in the caption or title)

b) in column 5 use a short acronym for “Visible-light irradiation”

c) remove the unity of measure from the cells on column 2-4 and report it only in the titles of the columns

8) Fig 11 a,b. Change color of LHD spectra. It is not visible after printing. Number the spectra as in fig 2 and 4.

9) some mistakes to correct:

line 58, 59, 60   usd

line 124, 126 … hydroxy

line 155… oxidbillity

Author Response

Dear reviewer 2,

Thank you very much for reviewing our manuscript, and finding out so detailed  mistakes which I think actually can improve the quality of our paper. I am also particularly grateful to you for your acknowledgement on the novelty of our paper. Now, I have revised our manuscript according to your comments, all the revisions are highlighted in red, and the detailed revision processes are described as follows:

Comment 1:

The data are clearly exposed, but authors may improve the clarity of paper modifying figures and tables basing on the following minor revisions.

Response to reviewer 2:

Thank you very much for your advice on modifying the figures and tables in our manuscript. All the figures and tables have been revised based on the following minor revisions.

Comment 2:

ENGLISH: some misprints are present. A in-depth revision of the language is needed.

Response to reviewer 2:

I am sorry for my carelessness. Now, I have taken a in-depth revision of the language for our manuscript.

Comment 3:

Minor revisions:

1) CuFe-LHD is defined also as the simpler LHD in the junction ND/LHD, or in the figures; whereas in the text and in the table1 was reported as CuFe-LHD or LDH. This can generate confusion. I suggest to always use the same form in the test, figures and table: always LHD or always CuFe-LHD.

Response to reviewer 2:

Thank you very much for your advice. Now, CuFe-LDH is replaced by LDH uniformly in the revised manuscript. I think it will not generate confusion again.

2) line 85-87 “The crystal phase and crystallinity of ND, CuFe-LDH and ND/LDH were characterized by using a Rigaku X-ray diffractometer with Cu Kα radiation over a range of 2θangles from 20° to 90° (Figure 2), respectively”

In experimental is reported a Shimatzu instrument. Is that a different XRD instrument? It should clarified and that information should be moved in the experimental.

Response to reviewer 2:

I am sorry for my carelessness. The ND, LDH and ND/LDH samples were characterized by a Rigaku X-ray diffractometer firstly, but the testing results were not satisfied. Then, we used another X-ray diffractometer (XRD-6000, Shimadzu) to characterize the samples, and the testing results were used in our paper. So, I confused these two X-ray diffractometers. Now, I have deleted “Rigaku” in the revised manuscript.

3) line 127-129 “The peak at cm-1 was ascribed to the stretching vibration of C-O structure which was subject to carbonate in the anionic interlayer of CuFe- LDH or oxygen-containing group on ND surface”

This sentence is not clear at all. Authors should better write and better explain.

Response to reviewer 2:

Thank you very much for this comment. This sentence has been revised as follow: The peak at 1385 cm-1 was ascribed to the stretching vibration of C-O structure which was from the carbonate located in the anionic interlayer of LDH, or from the oxygen-containing group on ND surface.

4) Fig 1 (A,B,C,D). letters ABCD in blue color are not visible in black and white printed version. In addition, they are to much small: they have to be increased in dimensions. The lattice fringes and the numbers in red color are not visible in black and white printed version. I suggest to change color in order to make the text legible even in the printed version.

Response to reviewer 2:

Thank you very much for your advice. The letters ABCD are amplified. All the letters, texts, drawing lines and arrows are in white or black. I think the revised Figure 1 is clear in black and white printed version.

5) Fig. 2  I suggest to number the three XRD spectra in order to recognize even in the printed version in black and white, and add the corresponding information in the caption.

Response to reviewer 2:

Thank you very much for your advice. The three XRD spectra in Figure 2 have been labeled as A, B and C for ND, LDH and ND/LDH, respectively. The corresponding information are also added in the caption of Figure 2.

6) Fig. 4  I suggest to number the two FTIR spectra in order to recognize even in the printed version in black and white, and add the corresponding information in the caption.

Response to reviewer 2:

Thank you very much for your advice. The three XRD spectra in Figure 4 have been labeled as A and B for LDH and ND/LDH, respectively. The corresponding information are also added in the caption of Figure 4.

7) Table 1. The table is not clearly readable due to much text in the cells. I suggest:

a) eliminate the column 3 that reports the same information for all sample( move info in the caption or title)

b) in column 5 use a short acronym for “Visible-light irradiation”

c) remove the unity of measure from the cells on column 2-4 and report it only in the titles of the columns

Response to reviewer 2:

Thank you very much for your advice. The column 3 has been deleted. A short acronym “VLI” is used to replace “Visible-light irradiation” in the column of “Irradiation condition”. These two pieces of information have been added as a note under the table 1. The unity (g/L) for the dosage of LDH or ND/LDH is removed only in the title of the column. In the column of “Dosage of radical scavenger”, the concentration and the kind of radical scavenger are all different, so I think the current format is OK.

8) Fig 11 a,b. Change color of LHD spectra. It is not visible after printing. Number the spectra as in fig 2 and 4.

Response to reviewer 2:

Thank you very much for your advice. The color of LDH spectra in Figure 11 have been revised into black. LDH and ND/LDH have been labeled as A and B, respectively. The corresponding information are also added in the caption of Figure 11.

9) some mistakes to correct:

line 58, 59, 60   usd

line 124, 126 … hydroxy

line 155… oxidbillity

Response to reviewer 2:

I am sorry for my carelessness. “usd” has been replaced by “used”. “hydroxy” has been replaced by “hydroxyl”. “oxidbillity” has been replaced by “oxidability”. In addition, I have carefully checked all the English language of our manuscript.

Thank you very much again for your comments. I hope this revision is satisfied to you and our paper could be accepted soon.

Round  2

Reviewer 1 Report

The revision was performed well. Now this paper is suitable for publishing in Catalysts journal.